# Online Learning for Adversaries with Memory: Price of Past Mistakes

**Oren Anava**
Technion
Haifa, Israel
oanava@tx.technion.ac.il

**Elad Hazan**
Princeton University
New York, USA
ehazan@cs.princeton.edu

**Shie Mannor**
Technion
Haifa, Israel
shie@ee.technion.ac.il

## Abstract

The framework of online learning with memory naturally captures learning problems with temporal effects, and was previously studied for the experts setting. In this work we extend the notion of learning with memory to the general Online Convex Optimization (OCO) framework, and present two algorithms that attain low regret. The first algorithm applies to Lipschitz continuous loss functions, obtaining optimal regret bounds for both convex and strongly convex losses. The second algorithm attains the optimal regret bounds and applies more broadly to convex losses without requiring Lipschitz continuity, yet is more complicated to implement. We complement the theoretical results with two applications: statistical arbitrage in finance, and multi-step ahead prediction in statistics.

## 1 Introduction

Online learning is a well-established learning paradigm which has both theoretical and practical appeals. The goal in this paradigm is to make a sequential decision, where at each trial the cost associated with previous prediction tasks is given. In recent years, online learning has been widely applied to several research fields including game theory, information theory, and optimization. We refer the reader to [1, 2, 3] for more comprehensive survey.

One of the most well-studied frameworks of online learning is *Online Convex Optimization* (OCO). In this framework, an online player iteratively chooses a decision in a convex set, then a convex loss function is revealed, and the player suffers loss that is the convex function applied to the decision she chose. It is usually assumed that the loss functions are chosen arbitrarily, possibly by an all-powerful adversary. The performance of the online player is measured using the *regret* criterion, which compares the accumulated loss of the player with the accumulated loss of the best fixed decision in hindsight.

The above notion of regret captures only *memoryless adversaries* who determine the loss based on the player's current decision, and fails to cope with *bounded-memory adversaries* who determine the loss based on the player's current and previous decisions. However, in many scenarios such as coding, compression, portfolio selection and more, the adversary is not completely memoryless and the previous decisions of the player affect her current loss. We are particularly concerned with scenarios in which the memory is relatively short-term and simple, in contrast to state-action models for which reinforcement learning models are more suitable [4].

An important aspect of our work is that the memory is *not* used to relax the adaptiveness of the adversary (cf. [5, 6]), but rather to model the feedback received by the player. In particular, throughout this work we assume that the adversary is *oblivious*, that is, must determine the whole set of loss functions in advance. In addition, we assume a *counterfactual feedback* model: the player is aware of the loss she would suffer had she played any sequence of $m$ decisions in the previous $m$ rounds. This model is quite common in the online learning literature; see for instance [7, 8].

Our goal in this work is to extend the notion of learning with memory to one of the most general online learning frameworks - the OCO. To this end, we adapt the *policy regret*[1] criterion of [5], and propose two different approaches for the extended framework, both attain the optimal bounds with respect to this criterion.

## 1.1 Summary of Results

We present and analyze two algorithms for the framework of OCO with memory, both attain policy regret bounds that are optimal in the number of rounds. Our first algorithm utilizes the Lipschitz property of the loss functions, and — to the best of our knowledge — is the first algorithm for this framework that is not based on any blocking technique (this technique is detailed in the related work section below). This algorithm attains $\mathcal{O}(T^{1/2})$-policy regret for general convex loss functions and $\mathcal{O}(\log T)$-policy regret for strongly convex losses.

For the case of convex and non-Lipschitz loss functions, our second algorithm attains the nearly optimal $\tilde{\mathcal{O}}(T^{1/2})$-policy regret[2]; its downside is that it is randomized and more difficult to implement. A novel result that follows immediately from our analysis is that *our second algorithm attains an expected $\tilde{\mathcal{O}}(T^{1/2})$-regret, along with $\tilde{\mathcal{O}}(T^{1/2})$ decision switches in the standard OCO framework*. Similar result currently exists only for the special case of the experts problem [9]. We note that the two algorithms we present are related in spirit (both designed to cope with bounded-memory adversaries), but differ in the techniques and analysis.

| Framework | Previous bound | Our first approach | Our second approach |
|---|---|---|---|
| Experts with Memory | $\mathcal{O}(T^{1/2})$ | Not applicable | $\tilde{\mathcal{O}}(T^{1/2})$ |
| OCO with memory (convex losses) | $\mathcal{O}(T^{2/3})$ | $\mathcal{O}(T^{1/2})$ | $\tilde{\mathcal{O}}(T^{1/2})$ |
| OCO with Memory (strongly convex losses) | $\tilde{\mathcal{O}}(T^{1/3})$ | $\mathcal{O}(\log T)$ | $\tilde{\mathcal{O}}(T^{1/2})$ |

Table 1: State-of-the-art upper-bounds on the policy regret as a function of $T$ (number of rounds) for the framework of OCO with memory. The best known bounds are due to the works of [9], [8], and [5], which are detailed in the related work section below.

## 1.2 Related Work

The framework of OCO with memory was initially considered in [7] as an extension to the experts framework of [10]. *Merhav et al.* offered a blocking technique that guarantees a policy regret bound of $\mathcal{O}(T^{2/3})$ against bounded-memory adversaries. Roughly speaking, the proposed technique divides the $T$ rounds into $T^{2/3}$ equal-sized blocks, while employing a constant decision throughout each of these blocks. The small number of decision switches enables the learning in the extended framework, yet the constant block size results in a suboptimal policy regret bound.

Later, [8] showed that a policy regret bound of $\mathcal{O}(T^{1/2})$ can be achieved by simply adapting the Shrinking Dartboard (SD) algorithm of [9] to the framework considered in [7]. In short, the SD algorithm is aimed at ensuring an expected $\mathcal{O}(T^{1/2})$ decision switches in addition to $\mathcal{O}(T^{1/2})$-regret. These two properties together enable the learning in the considered framework, and the randomized block size yields an optimal policy regret bound. Note that in both [7] and [8], the

presented techniques are applicable only to the variant of the experts framework to adversaries with memory, and not to the general OCO framework.

The framework of online learning against adversaries with memory was studied also in the setting of the adversarial multi-armed bandit problem. In this context, [5] showed how to convert an online learning algorithm with regret guarantee of $\mathcal{O}(T^q)$ into an online learning algorithm that attains $\mathcal{O}(T^{1/(2-q)})$-policy regret, also using a blocking technique. This approach is in fact a generalization of [7] to the bandit setting, yet the ideas presented are somewhat simpler. Despite the original presentation of [5] in the bandit setting, their ideas can be easily generalized to the framework of OCO with memory, yielding a policy regret bound of $\mathcal{O}(T^{2/3})$ for convex losses and $\tilde{\mathcal{O}}(T^{1/3})$-policy regret for strongly convex losses.

An important concept that is captured by the framework of OCO with memory is switching costs, which can be seen as a special case where the memory is of length 1. This special case was studied in the works of [11], who studied the relationship between second order regret bounds and switching costs; and [12], who proved that the blocking algorithm of [5] is optimal for the setting of the adversarial multi-armed bandit with switching costs.

## 2 Preliminaries and Model

We continue to formally define the notation for both the standard OCO framework and the framework of OCO with memory. For sake of readability, we shall use the notation $g_t$ for memoryless loss functions (that correspond to memoryless adversaries), and $f_t$ for loss functions with memory (that correspond to bounded-memory adversaries).

### 2.1 The Standard OCO Framework

In the standard OCO framework, an online player iteratively chooses a decision $x_t \in \mathcal{K}$, and suffers loss that is equal to $g_t(x_t)$. The decision set $\mathcal{K}$ is assumed to be a bounded convex subset of $\mathbb{R}^n$, and the loss functions $\{g_t\}_{t=1}^T$ are assumed to be convex functions from $\mathcal{K}$ to $[0, 1]$. In addition, the set $\{g_t\}_{t=1}^T$ is assumed to be chosen in advance, possibly by an all-powerful adversary that has full knowledge of our learning algorithm (see [1], for instance). The performance of the player is measured using the *regret* criterion, defined as follows:

$$R_T = \sum_{t=1}^T g_t(x_t) - \min_{x \in \mathcal{K}} \sum_{t=1}^T g_t(x),$$

where $T$ is a predefined integer denoting the total number of rounds played. The goal in this framework is to design efficient algorithms, whose regret grows sublinearly in $T$, corresponding to an average per-round regret going to zero as $T$ increases.

### 2.2 The Framework of OCO with Memory

In this work we consider the framework of OCO with memory, detailed as follows: at each round $t$, the online player chooses a decision $x_t \in \mathcal{K} \subset \mathbb{R}^n$. Then, a loss function $f_t : \mathcal{K}^{m+1} \to \mathbb{R}$ is revealed, and the player suffers loss of $f_t(x_{t-m}, \ldots, x_t)$. For simplicity, we assume that $0 \in \mathcal{K}$, and that $f_t(x_0, \ldots, x_m) \in [0, 1]$ for any $x_0, \ldots, x_m \in \mathcal{K}$. Notice that the loss at round $t$ depends on the previous $m$ decisions of the player, as well as on his current one. We assume that after $f_t$ is revealed, the player is aware of the loss she would suffer had she played any sequence of decisions $x_{t-m}, \ldots, x_t$ (this corresponds to the counterfactual feedback model mentioned earlier).

Our goal in this framework is to minimize the *policy regret*, as defined in [5][3]:

$$R_{T,m} = \sum_{t=m}^T f_t(x_{t-m}, \ldots, x_t) - \min_{x \in \mathcal{K}} \sum_{t=m}^T f_t(x, \ldots, x).$$

We define the notion of convexity for the loss functions $\{f_t\}_{t=1}^T$ as follows: we say that $f_t$ is a convex loss function with memory if $\tilde{f}_t(x) = f_t(x, \ldots, x)$ is convex in $x$. From now on, we assume that

**Algorithm 1**

---
1: Input: learning rate $\eta > 0$, $\sigma$-strongly convex and smooth regularization function $\mathcal{R}(x)$.
2: Choose $x_0, \ldots, x_m \in \mathcal{K}$ arbitrarily.
3: **for** $t = m$ to $T$ **do**
4:     Play $x_t$ and suffer loss $f_t(x_{t-m}, \ldots, x_t)$.
5:     Set $x_{t+1} = \arg\min_{x \in \mathcal{K}} \left\{ \eta \cdot \sum_{\tau=m}^{t} \tilde{f}_\tau(x) + \mathcal{R}(x) \right\}$
6: **end for**

---

$\{f_t\}_{t=1}^T$ are convex loss functions with memory. This assumption is necessary in some cases, if *efficient* algorithms are considered; otherwise, the optimization problem $\min_{x \in \mathcal{K}} \sum_{t=m}^T f_t(x, \ldots, x)$ might not be solvable efficiently.

## 3   Policy Regret for Lipschitz Continuous Loss Functions

In this section we assume that the loss functions $\{f_t\}_{t=1}^T$ are Lipschitz continuous for some Lipschitz constant $L$, that is

$$|f_t(x_0, \ldots, x_m) - f_t(y_0, \ldots, y_m)| \le L \cdot \|(x_0, \ldots, x_m) - (y_0, \ldots, y_m)\|,$$

and adapt the well-known Regularized Follow The Leader (RFTL) algorithm to cope with bounded-memory adversaries. In the above and throughout the paper, we use $\| \cdot \|$ to denote the $\ell_2$-norm. Due to space constraints we present here only the algorithm and the main theorem, and defer the complete analysis to the supplementary material.

Intuitively, Algorithm 1 relies on the fact that the corresponding functions $\{\tilde{f}_t\}_{t=1}^T$ are memoryless and convex. Thus, standard regret minimization techniques are applicable, yielding a regret bound of $\mathcal{O}(T^{1/2})$ for $\{\tilde{f}_t\}_{t=1}^T$. This however, is not the policy regret bound we are interested in, but is in fact quite close if we use the Lipschitz property of $\{f_t\}_{t=1}^T$ and set the learning rate properly. The algorithm requires the following standard definitions of $R$ and $\lambda$ (see supplementary material for more comprehensive background and exact norm definitions):

$$\lambda = \sup_{t \in \{1, \ldots, T\}, x, y \in \mathcal{K}} \left\{ \left( \|\nabla \tilde{f}_t(x)\|_y^* \right)^2 \right\} \quad \text{and} \quad R = \sup_{x, y \in \mathcal{K}} \{\mathcal{R}(x) - \mathcal{R}(y)\}. \tag{1}$$

Additionally, we denote by $\sigma$ the strong convexity[4] parameter of the regularization function $\mathcal{R}(x)$.

For Algorithm 1 we can prove the following:

**Theorem 3.1.** *Let $\{f_t\}_{t=1}^T$ be Lipschitz continuous loss functions with memory (from $\mathcal{K}^{m+1}$ to $[0, 1]$), and let $R$ and $\lambda$ be as defined in Equation (1). Then, Algorithm 1 generates an online sequence $\{x_t\}_{t=1}^T$, for which the following holds:*

$$R_{T,m} = \sum_{t=m}^T f_t(x_{t-m}, \ldots, x_t) - \min_{x \in \mathcal{K}} \sum_{t=m}^T f_t(x, \ldots, x) \le 2T\lambda\eta(m+1)^{3/2} + \frac{R}{\eta}.$$

*Setting $\eta = R^{1/2}(TL)^{-1/2}(m+1)^{-3/4}(\lambda/\sigma)^{-1/4}$ yields $R_{T,m} \le 3(TRL)^{1/2}(m+1)^{3/4}(\lambda/\sigma)^{1/4}$.*

The following is an immediate corollary of Theorem 3.1 to $H$-strongly convex losses:

**Corollary 3.2.** *Let $\{f_t\}_{t=1}^T$ be Lipschitz continuous and $H$-strongly convex loss functions with memory (from $\mathcal{K}^{m+1}$ to $[0, 1]$), and denote $G = \sup_{t, x \in \mathcal{K}} \|\nabla \tilde{f}_t(x)\|$. Then, Algorithm 1 generates an online sequence $\{x_t\}_{t=1}^T$, for which the following holds:*

$$R_{T,m} \le 2(m+1)^{3/2} G^2 \sum_{t=m}^T \eta_t + \sum_{t=m}^T \|x_t - x^*\|^2 \left( \frac{1}{\eta_{t+1}} - \frac{1}{\eta_t} - H \right).$$

*Setting $\eta_t = \frac{1}{Ht}$ yields $R_{T,m} \le \frac{2(m+1)^{3/2} G^2}{H} (1 + \log(T))$.*

The proof simply requires plugging time-dependent learning rate in the proof of Theorem 3.1, and is thus omitted here.

**Algorithm 2**

1: Input: learning parameter $\eta > 0$.
2: Initialize $w_1(x) = 1$ for all $x \in \mathcal{K}$, and choose $x_1 \in \mathcal{K}$ arbitrarily.
3: **for** $t = 1$ to $T$ **do**
4:    Play $x_t$ and suffer loss $g_t(x_t)$.
5:    Define weights $w_{t+1}(x) = e^{-\alpha \sum_{\tau=1}^{t} \hat{g}_\tau(x)}$, where $\alpha = \frac{\eta}{4G^2}$ and $\hat{g}_t(x) = g_t(x) + \frac{\eta}{2}\|x\|^2$.
6:    Set $x_{t+1} = x_t$ with probability $\frac{w_{t+1}(x_t)}{w_t(x_t)}$.
7:    Otherwise, sample $x_{t+1}$ from the density function $p_{t+1}(x) = w_{t+1}(x) \cdot \left( \int_\mathcal{K} w_{t+1}(x)dx \right)^{-1}$.
8: **end for**

## 4 Policy Regret with Low Switches

In this section we present a different approach to the framework of OCO with memory — low switches. This approach was considered before in [8], who adapted the Shrinking Dartboard (SD) algorithm of [9] to cope with limited-delay coding. However, the authors in [9, 8] consider only the experts setting, in which the decision set is the simplex and the loss functions are linear. Here we adapt this approach to general decision sets and generally convex loss functions, and obtain optimal policy regret against bounded-memory adversaries.

Due to space constraints, we present here only the algorithm and main theorem. The complete analysis appears in the supplementary material.

Intuitively, Algorithm 2 defines a probability distribution over $\mathcal{K}$ at each round $t$. By sampling from this probability distribution one can generate an online sequence that has an expected low regret guarantee. This however is not sufficient in order to cope with bounded-memory adversaries, and thus an additional element of choosing $x_{t+1} = x_t$ with high probability is necessary (line 6). Our analysis shows that if this probability is equal to $\frac{w_{t+1}(x_t)}{w_t(x_t)}$ the regret guarantee remains, and we get an additional low switches guarantee.

For Algorithm 2 we can prove the following:

**Theorem 4.1.** *Let $\{g_t\}_{t=1}^T$ be convex functions from $\mathcal{K}$ to $[0,1]$, such that $D = \sup_{x,y \in \mathcal{K}} \|x - y\|$ and $G = \sup_{x,t} \|\nabla g_t(x)\|$, and define $\hat{g}_t(x) = g_t(x) + \frac{\eta}{2}\|x\|^2$ for $\eta = \frac{2G}{D}\sqrt{\frac{1+\log(T+1)}{T}}$. Then, Algorithm 2 generates an online sequence $\{x_t\}_{t=1}^T$, for which it holds that*

$$\mathbb{E}[R_T] = \mathcal{O}\big(\sqrt{T\log(T)}\big) \quad \text{and} \quad \mathbb{E}[S] = \mathcal{O}\big(\sqrt{T\log(T)}\big),$$

*where $S$ is the number of decision switches in the sequence $\{x_t\}_{t=1}^T$.*

The exact bounds for $\mathbb{E}[R_T]$ and $\mathbb{E}[S]$ are given in the supplementary material. Notice that Algorithm 2 applies to memoryless loss functions, yet its low switches guarantee implies learning against bounded-memory adversaries as stated and proven in Lemma C.5 (see supplementary material).

## 5 Application to Statistical Arbitrage

Our first application is motivated by financial models that are aimed at creating statistical arbitrage opportunities. In the literature, "statistical arbitrage" refers to statistical mispricing of one or more assets based on their expected value. One of the most common trading strategies, known as "pairs trading", seeks to create a mean reverting portfolio using two assets with same sectorial belonging (typically using both long and short sales). Then, by buying this portfolio below its mean and selling it above, one can have an expected positive profit with low risk.

Here we extend the traditional pairs trading strategy, and present an approach that aims at constructing a mean reverting portfolio from an arbitrary (yet known in advance) number of assets. Roughly speaking, our goal is to synthetically create a mean reverting portfolio by maintaining weights upon $n$ different assets. The main problem arises in this context is how do we quantify the amount of mean reversion of a given portfolio? Indeed, mean reversion is somewhat an ill-defined concept, and thus

different proxies are usually defined to capture its notion. We refer the reader to [13, 14, 15], in which few of these proxies (such as predictability and zero-crossing) are presented.

In this work, we consider a proxy that is aimed at preserving the mean price of the constructed portfolio (over the last $m$ trading periods) close to zero, while maximizing its variance. We note that due to the very nature of the problem (weights of one trading period affect future performance), the memory comes unavoidably into the picture.

We proceed to formally define the new mean reversion proxy and the use of our new algorithm in this model. Thus, denote by $y_t \in \mathbb{R}^n$ the prices of $n$ assets at time $t$, and by $x_t \in \mathbb{R}^n$ a distribution of weights over these assets. Since short selling is allowed, the norm of $x_t$ can sum up to an arbitrary number, determined by the loan flexibility. Without loss of generality we assume that $\|x_t\|_2 = 1$, which is also assumed in the works of [14, 15]. Note that since $x_t$ determines the proportion of wealth to be invested in each asset and not the actual wealth it self, any other constant would work as well. Consequently, define:

$$f_t(x_{t-m}, \ldots, x_t) = \left( \sum_{i=0}^{m} x_{t-i}^\top y_{t-i} \right)^2 - \lambda \cdot \sum_{i=0}^{m} \left( x_{t-i}^\top y_{t-i} \right)^2, \tag{2}$$

for some $\lambda > 0$. Notice that minimizing $f_t$ iteratively yields a process $\{x_t^\top y_t\}_{t=1}^T$ such that its mean is close to zero (due to the expression on the left), and its variance is maximized (due to the expression on the right). We use the regret criterion to measure our performance against the best distribution of weights in hindsight, and wish to generate a series of weights $\{x_t\}_{t=1}^T$ such that the regret is sublinear. Thus, define the memoryless loss function $\tilde{f}_t(x) = f_t(x, \ldots, x)$ and denote

$$A_t = \sum_{i=0}^{m-1} \sum_{j=0}^{m-1} y_{t-i} y_{t-j}^\top \quad \text{and} \quad B_t = \lambda \cdot \left( \sum_{i=0}^{m-1} y_{t-i} y_{t-i}^\top \right).$$

Notice we can write $\tilde{f}_t(x) = x^\top A_t x - x^\top B_t x$. Since $\tilde{f}_t$ is not convex in general, our techniques are not straightforwardly applicable here. However, the hidden convexity of the problem allows us to bypass this issue by a simple and tight Positive Semi-Definite (PSD) relaxation. Define

$$h_t(X) = X \circ A_t - X \circ B_t, \tag{3}$$

where $X$ is a PSD matrix with $Tr(X) = 1$, and $X \circ A$ is defined as $\sum_{i=1}^n \sum_{j=1}^n X(i,j) \cdot A(i,j)$. Now, notice that the problem of minimizing $\sum_{t=m}^T h_t(X)$ is a PSD relaxation to the minimization problem $\sum_{t=m}^T \tilde{f}_t(x)$, and for the optimal solution it holds that:

$$\min_X \sum_{t=m}^T h_t(X) \le \sum_{t=m}^T h_t(x^* x^{*\top}) = \sum_{t=m}^T \tilde{f}_t(x^*).$$

where $x^* = \arg\min_{x \in \mathcal{K}} \sum_{t=m}^T \tilde{f}_t(x)$. Also, we can recover a vector $x$ from the PSD matrix $X$ using an eigenvector decomposition as follows: represent $X = \sum_{i=1}^n \lambda_i v_i v_i^\top$, where each $v_i$ is a unit vector and $\lambda_i$ are non-negative coefficients such that $\sum_{i=1}^n \lambda_i = 1$. Then, by sampling the eigenvector $x = v_i$ with probability $\lambda_i$, we get that $\mathbb{E}[\tilde{f}_t(x)] = h_t(X)$. Technically, this decomposition is possible due to the fact that $X$ is a PSD matrix with $Tr(X) = 1$. Notice that $h_t$ is linear in $X$, and thus we can apply regret minimization techniques on the loss functions $\{h_t\}_{t=1}^T$. This procedure is formally given in Algorithm 3. For this algorithm we can prove the following:

**Corollary 5.1.** *Let $\{f_t\}_{t=1}^T$ be as defined in Equation (2), and $\{h_t\}_{t=1}^T$ be the corresponding memoryless functions, as defined in Equation (3). Then, applying Algorithm 2 to the loss functions $\{h_t\}_{t=1}^T$ yields an online sequence $\{X_t\}_{t=1}^T$, for which the following holds:*

$$\sum_{t=1}^T \mathbb{E}\left[h_t(X_t)\right] - \min_{\substack{X \succeq 0 \\ Tr(X)=1}} \sum_{t=1}^T h_t(X) = \mathcal{O}\big(\sqrt{T \log(T)}\big).$$

*Sampling $x_t \sim X_t$ using the eigenvector decomposition described above yields:*

$$\mathbb{E}\left[R_{T,m}\right] = \sum_{t=m}^T \mathbb{E}\left[f_t(x_{t-m}, \ldots, x_t)\right] - \min_{\|x\|=1} \sum_{t=m}^T f_t(x, \ldots, x) = \mathcal{O}\big(\sqrt{T \log(T)}\big).$$

---

**Algorithm 3** Online Statistical Arbitrage (OSA)

---

1: Input: Learning rate $\eta$, memory parameter $m$, regularizer $\lambda$.
2: Initialize $X_1 = \frac{1}{n} I_{n \times n}$.
3: **for** $t = 1$ to $T$ **do**
4:    Randomize $x_t \sim X_t$ using the eigenvector decomposition.
5:    Observe $f_t$ and define $h_t$ as in equation (3).
6:    Apply Algorithm 2 to $h_t(X_t)$ to get $X_{t+1}$.
7: **end for**

---

**Remark**: We assume here that the prices of the $n$ assets at round $t$ are bounded for all $t$ by a constant which is independent of $T$.

The main novelty of our approach to the task of constructing mean reverting portfolios is the ability to maintain the weight distributions online. This is in contrast to the traditional offline approaches that require a training period (to learn a weight distribution), and a trading period (to apply a corresponding trading strategy).

# 6 Application to Multi-Step Ahead Prediction

Our second application is motivated by statistical models for time series prediction, and in particular by statistical models for multi-step ahead AR prediction. Thus, let $\{X_t\}_{t=1}^T$ be a time series (that is, a series of signal observations). The traditional AR (short for autoregressive) model, parameterized by lag $p$ and coefficient vector $\alpha \in \mathbb{R}^p$, assumes that each observation complies with

$$X_t = \sum_{k=1}^p \alpha_k X_{t-k} + \epsilon_t,$$

where $\{\epsilon_t\}_{t \in \mathbb{Z}}$ is white noise. In words, the model assumes that $X_t$ is a noisy linear combination of the previous $p$ observations. Sometimes, an additional additive term $\alpha_0$ is included to indicate drift, but we ignore this for simplicity.

The online setting for time series prediction is well-established by now, and appears in the works of [16, 17]. Here, we adapt this setting to the task of multi-step ahead AR prediction as follows: at round $t$, the online player has to predict $X_{t+m}$, while at her disposal are all the previous observations $X_1, \ldots, X_{t-1}$ (the parameter $m$ determines the number of steps ahead). Then, $X_t$ is revealed and she suffers loss of $f_t(X_t, \tilde{X}_t)$, where $\tilde{X}_t$ denotes her prediction for $X_t$. For simplicity, we consider the squared loss to be our error measure, that is, $f_t(X_t, \tilde{X}_t) = (X_t - \tilde{X}_t)^2$.

In the statistical literature, a common approach to the problem of multi-step ahead prediction is to consider 1-step ahead recursive AR predictors [18, 19]: essentially, this approach makes use of standard methods (e.g., maximum likelihood or least squares estimation) to extract the 1-step ahead estimator. For instance, a least squares estimator for $\alpha$ at round $t$ would be:

$$\alpha^{\mathrm{LS}} = \arg\min_\alpha \left\{ \sum_{\tau=1}^{t-1} \left( X_\tau - \tilde{X}_\tau^{\mathrm{AR}}(\alpha) \right)^2 \right\} = \arg\min_\alpha \left\{ \sum_{\tau=1}^{t-1} \left( X_\tau - \sum_{k=1}^p \alpha_k X_{\tau-k} \right)^2 \right\}.$$

Then, $\alpha^{\mathrm{LS}}$ is used to generate a prediction for $X_t$: $\tilde{X}_t^{\mathrm{AR}}(\alpha^{\mathrm{LS}}) = \sum_{i=1}^p \alpha_i^{\mathrm{LS}} X_{t-i}$, which is in turn used as a proxy for it in order to predict the value of $X_{t+1}$:

$$\tilde{X}_{t+1}^{\mathrm{AR}}(\alpha^{\mathrm{LS}}) = \alpha_1^{\mathrm{LS}} \tilde{X}_t^{\mathrm{AR}}(\alpha^{\mathrm{LS}}) + \sum_{k=2}^p \alpha_k^{\mathrm{LS}} X_{t-k+1}. \tag{4}$$

The values of $X_{t+2}, \ldots, X_{t+m}$ are predicted in the same recursive manner. The most obvious drawback of this approach is that not much can be said on the quality of this predictor even if the AR model is well-specified, let alone if it is not (see [18] for further discussion on this issue).

In light of this, the motivation to formulate the problem of multi-step ahead prediction in the online setting is quite clear: attaining regret in this setting would imply that our algorithm's performance

---

**Algorithm 4** Adaptation of Algorithm 1 to Multi-Step Ahead Prediction

---
1: Input: learning rate $\eta$, regularization function $\mathcal{R}(x)$, signal $\{X_t\}_{t=1}^T$.
2: Choose $w^0, \dots, w^m \in \mathcal{K}^{\mathrm{IP}}$ arbitrarily.
3: **for** $t = m$ to $T$ **do**
4:     Predict $\tilde{X}_t^{\mathrm{IP}}(w^{t-m}) = \sum_{k=1}^p w_k^{t-m} X_{t-m-k}$ and suffer loss $\left(X_t - \tilde{X}_t^{\mathrm{IP}}(w^{t-m})\right)^2$.
5:     Set $w^{t+1} = \arg\min_{w \in \mathcal{K}^{\mathrm{IP}}} \left\{ \eta \sum_{\tau=m}^t \left(X_\tau - \tilde{X}_\tau^{\mathrm{IP}}(w)\right)^2 + \|w\|_2^2 \right\}$
6: **end for**

---

is comparable with the best 1-step ahead recursive AR predictor in hindsight (even if the latter is misspecified). Thus, our goal is to minimize the following regret term:

$$R_T = \sum_{t=1}^T \left(X_t - \tilde{X}_t\right)^2 - \min_{\alpha \in \mathcal{K}} \sum_{t=1}^T \left(X_t - \tilde{X}_t^{\mathrm{AR}}(\alpha)\right)^2,$$

where $\mathcal{K}$ denotes the set of all 1-step ahead recursive AR predictors, against which we want to compete. Note that since the feedback is delayed (the AR coefficients chosen at round $t-m$ are used to generate the prediction at round $t$), the memory comes unavoidably into the picture. Nevertheless, here also both of our techniques are not straightforwardly applicable due the non-convex structure of the problem: each prediction $\tilde{X}_t^{\mathrm{AR}}(\alpha)$ contains products of $\alpha$ coefficients that cause the losses to be non-convex in $\alpha$.

To circumvent this issue, we use non-proper learning techniques, and let our predictions to be of the form $\tilde{X}_{t+m}^{\mathrm{NP}}(w) = \sum_{k=1}^p w_k X_{t-k}$ for a properly chosen set $\mathcal{K}^{\mathrm{NP}} \subset \mathbb{R}^p$ of the $w$ coefficients. Basically, the idea is to show that (a) attaining regret bound with respect to the best predictor in the new family can be done using the techniques we present in this work; and (b) the best predictor in the new family is better than the best 1-step ahead recursive AR predictor. This would imply a regret bound with respect to best 1-step ahead recursive AR predictor in hindsight. Our formal result is given in the following corollary:

**Corollary 6.1.** *Let $D = \sup_{w_1, w_2 \in \mathcal{K}^{IP}} \|w_1 - w_2\|_2$ and $G = \sup_{w,t} \|\nabla f_t(X_t, \tilde{X}_t(w))\|_2$. Then, Algorithm 4 generates an online sequence $\{w^t\}_{t=1}^T$, for which it holds that*

$$\sum_{t=1}^T \left(X_t - \tilde{X}_t^{IP}(w^{t-m})\right)^2 - \min_{\alpha \in \mathcal{K}} \sum_{t=1}^T \left(X_t - \tilde{X}_t^{AR}(\alpha)\right)^2 \leq 3GD\sqrt{Tm}.$$

**Remark**: The tighter bound in $m$ ($m^{1/2}$ instead of $m^{3/4}$) follows directly by modifying the proof of theorem 3.1 to this setting ($f_t$ is affected only by $w^{t-m}$ and not by $w^{t-m}, \dots, w^t$).

In the above, the values of $D$ and $G$ are determined by the choice of the set $\mathcal{K}$. For instance, if we want to compete against the best $\alpha \in \mathcal{K} = [-1, 1]^p$ we need to use the restriction $w_k \leq 2^m$ for all $k$. In this case, $D \approx 2^m$ and $G \approx 1$. If we consider $\mathcal{K}$ to be the set of all $\alpha \in \mathbb{R}^p$ such that $\alpha_k \leq (1/\sqrt{2})^k$, we get that $D \approx \sqrt{m}$ and $G \approx 1$.

The main novelty of our approach to the task of multi-step ahead prediction is the elimination of generative assumptions on the data, that is, we allow the time series to be arbitrarily generated. Such assumptions are common in the statistical literature, and needed in general to extract ML estimators.

## 7 Discussion and Conclusion

In this work we extended the notion of online learning with memory to capture the general OCO framework, and proposed two algorithms with tight regret guarantees. We then applied our algorithms to two extensively studied problems: construction of mean reverting portfolios, and multistep ahead prediction. It remains for future work to further investigate the performance of our algorithms in these problems and other problems in which the memory naturally arises.

### Acknowledgments

This work has been supported by the European Community's Seventh Framework Programme (FP7/2007-2013) under grant agreement 306638 (SUPREL).

## Footnotes

[1] The policy regret compares the performance of the online player with the best fixed sequence of actions in hindsight, and thus captures the notion of adversaries with memory. A formal definition appears in Section 2.

[2] The notation $\tilde{\mathcal{O}}(\cdot)$ is a variant of the $\mathcal{O}(\cdot)$ notation that ignores logarithmic factors.

[3]The rounds in which $t < m$ are ignored since we assume that the loss per round is bounded by a constant; this adds at most a constant to the final regret bound.

[4] $f(x)$ is $\sigma$-strongly convex if $\nabla^2 f(x) \succeq \sigma I_{n \times n}$ for all $x \in \mathcal{K}$. We say that $f_t : \mathcal{K}^{m+1} \to \mathbb{R}$ is $\sigma$-strongly convex loss function with memory if $\tilde{f}_t(x) = f_t(x, \ldots, x)$ is $\sigma$-strongly convex in $x$.

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
