[Supplementary Material]

## A    Experimental Results: Statistical Arbitrage

In this section we present some preliminary results that demonstrate the effectiveness of the proposed algorithm to the task of creating statistical arbitrage opportunities under the pairs trading setting. In this setting, we are given two assets with the same sectoral belonging and our goal is to construct a mean reverting portfolio by maintaining weights upon these assets. To simplify the setting we ignore transaction costs (both for our algorithm and the benchmarks).

In order to isolate the problem of constructing a mean reverting portfolio (which is of our interest) from the problem of designing a trading strategy, the experiments are executed in two stages: first, a mean reverting portfolio is constructed by each of the considered approaches (which are described below in Section A.1). Then, the same trading strategy is applied to all resulted portfolios, so that the different approaches are comparable in terms of return.

Our dataset contains time series of daily closing rates of 9 pairs of assets based on their common sectoral belonging (Coca Cola and Pepsi; AT&T and Verizon; Johnson&Johnson and Procter&Gamble; Cellcom and Partner; Microsoft and Intel; 3M and DD; Pfizer and Merck; Chevron and Exxon Mobil; and Home Depot and Wal-Mart). We use data between 01/01/2008 and 01/02/2013, which is divided into training set (75% of the data, from 01/01/2008 to 01/10/2011) and test set (25% of the data, from 02/10/2011 to 01/02/2013).

### A.1    Baselines

In order to capture the essence of our Online Statistical Arbitrage (OSA) algorithm with respect to its offline counterparts, we choose some of the fundamental offline approaches[5] to serve as benchmarks:

**Orthogonal Least Squares (OLS)** this baseline proposes to choose the eigenvector that corresponds to smallest eigenvalue of the empirical covariance matrix of $y_t$. This matrix is denoted by $C$, and formally defined as follows:

$$C = \frac{1}{T_{\text{training}} - 1} \cdot \sum_{t=1}^{T_{\text{training}}} \tilde{y}_t \tilde{y}_t^\top \quad , \quad \text{where} \quad \tilde{y}_t = y_t - \frac{1}{T_{\text{training}}} \cdot \sum_{t=1}^{T_{\text{training}}} y_t,$$

where $T_{\text{training}}$ denotes the number of days in the training set.

**Johansen Vector Error Correction Model** this baseline relies on co-integration techniques. Basically, co-integration is a statistical relationship where two time series (e.g., stock prices) that are both integrated of same order $d$ can be linearly combined to produce a single time series which is integrated of order $d - b$, where $b > 0$. In its application to pairs trading, the co-integration technique seeks to find a linear combination such that $d = b = 1$, which roughly results in a mean reverting combined asset.

**The offline optimum (Offline)** this baselines refers to the best distribution of weights in hindsight with respect to our proxy, that is

$$x_{\text{Offline}} = \arg\min_x \left\{ \sum_{t=m}^{T_{\text{test}}} \left( \sum_{i=0}^m x^T y_{t-i} \right)^2 - \lambda \cdot \sum_{i=0}^m \left( x^T y_{t-i} \right)^2 \right\}.$$

Here, $T_{\text{test}}$ denotes the number of days in the test set. Clearly, the performance of this baselines cannot be obtained in practice, as it relies on the future prices of the considered assets when constructing the portfolio. Nevertheless, this baseline has a crucial role in understanding the effectiveness of the proposed mean reversion proxy.

For the **OLS** and **Johansen** baselines we use the training period to generate a weight distribution $x$, and then construct the portfolio $\{x^\top y_t\}_{t=1}^{T_{\text{test}}}$. For **OSA** we run Algorithm 3 on the training set to get the sequence $\{x_t\}_{t=1}^{T_{\text{training}}}$. Then, we use $x_{T_{\text{training}}}$ as a warm start for a new run of Algorithm 3 on the test data to generate the portfolio $\{x_t^\top y_t\}_{t=1}^{T_{\text{test}}}$ (which will be used for the benchmark task).

Figure 1: **Sample experimental results of OSA and Johansen for the pair Coca Cola and Pepsi**

## A.2 Trading Strategy

In order to compare the different approaches, we apply the trading strategy of [22] to each of the resulting portfolios. Basically, [22] propose to take a position $N_t$ in the asset $z_t$ proportionally to $\frac{\alpha(\mu - z_t)}{\sigma^2} W_t$, where $W_t$ denotes the wealth at time $t$ and $\{z_t\}_{t=1}^{T}$ is assumed to be an auto regressive process of order 1 with mean $\mu$ that complies with $z_{t+1} = \alpha z_t + \sigma \epsilon_t$ (and $\epsilon_t \sim \mathcal{N}(0, 1)$). Essentially, this strategy takes a long position whenever the asset is below its mean and short position whenever it is above, while taking into account the autoregressive model parameters $\alpha$ and $\sigma$. In practice, these parameters are estimated on the training set and then used to generate $N_t$. A sample experiment for the pair Coca Cola and Pepsi (using the entire training and test sets) that compares the performance of our algorithm and Johansen's is illustrated in Figure 1.

## A.3 Results

In Figure 2 we plot the cumulative wealth of our online algorithm and the three offline baselines, and also provide the Sharpe ratios. To execute this experiments we use the 10 pairs of assets in our dataset. In all runs of our online algorithm and its offline counterpart we set $m = 5$ and $\lambda = 1$, arbitrarily. The task of determining the best values of $m$ and $\lambda$ is outside the scope of this paper, yet is a very challenging problem. The empirical observations clearly verify the effectiveness of the proposed mean reversion proxy and the online algorithm, as both **OSA** and **Offline** outperform the other baselines. It can can also be seen that the performance of **OSA** approaches the performance of **Offline** as time advances, corresponding to our theoretic regret guarantee. It remains for future work to compare the performance of the online approach and the offline state-of-the-art approaches in the presence of transaction costs.

| | Return (in %) | |
|---|---|---|
| | 8-month | 16-month |
| **Offline** | **39.45** | **102.67** |
| **OSA** | 33.59 | 98.33 |
| **OLS** | 23.64 | 83.68 |
| **Johansen** | 33.87 | 60.47 |

Figure 2: **Wealth as a function of time for the online algorithm and the three offline baselines**

# B  Experimental Results: Multi-Step Ahead Prediction

The following experiments demonstrate the effectiveness of the proposed algorithm under various synthetic settings. We start by presenting two state-of-the-art baselines for the problem at hand.

## B.1  Baselines

Most of the work on time series prediction considers what we call the *offline setting*: given a time series, compute the model parameters (in our case, the AR coefficients) and generate predictions for the signal accordingly. Our *online setting* can be seen as a sequential offline setting, in which at time $t$ we are given the time series values up to time $t-1$ and our task is to predict the signal value at time $t+m$. In light of this, we adapt the offline baselines presented below to the online setting. We note that this adaptation does not weaken the offline baselines in any way, and we use it only for comparison purposes.

**1-step ahead ML estimator.**  Essentially, this baseline aims to extract 1-step ahead estimator for $\alpha$ using maximum likelihood techniques. This estimator is then used to recursively predict the values of $X_t, \ldots, X_{t+m}$ as explained in Equation (4).

**Multi-step ahead ML estimator.**  This baselines extracts the multi-step ahead estimator using maximum likelihood techniques, and generates its predictions accordingly. See [18] for further discussion about this baseline and comparison with the previous one.

## B.2  Experimental Results

To evaluate the performance of our algorithms we design several various settings (presented below). In order to ensure the stability of the results we average them over 50 runs. In our tables, we mark with bold font the best results, and add an asterisk to indicate significance level of 0.05.

**Sanity check.**  We generate a time series that complies with the standard AR model using the coefficient vector $\alpha = [0.6, -0.5, 0.5, -0.4, 0.3]$ and i.i.d. noise terms that are distributed $\mathcal{N}(0, 0.3^2)$.

**AR mixture.**  Our motivation in this setting is to examine the functionality of the different algorithms when faced with abruptly changing environments. Thus, we consider a predefined set of AR coefficients, and generate time series by alternating between them in a random manner. We add an additive noise that is distributed $Uni[-0.5, 0.5]$.

**Slowly changing environment.**  Our motivation here is to test the robustness of our approach to slowly changing environment. Thus, we set

$$\alpha(t) = [-0.4, -0.5, 0.4, 0.4, 0.1] * \left(\frac{t}{10^4}\right) + [0.6, -0.4, 0.4, -0.5, 0.4] * \left(1 - \frac{t}{10^4}\right),$$

and generate the time series by adding uniformly distributed noise terms.

As evident in Figure **??** and Table **??**, our online algorithm outperforms the other algorithms when the time series exhibits some complicated structure. In the case where the error terms are Gaussian and the time series complies with the AR model (sanity check), we can see that all algorithms perform roughly the same, as can be expected by the theoretical guarantees of our algorithm and the MLE baselines.

## C Complete Analysis for Section 3

### C.1 Background

Recall the RFTL algorithm, which is one of the most popular algorithms for the standard OCO framework. Basically, RFTL generates the decision at round $t$ according to the following rule:

$$x_t = \arg\min_{x \in \mathcal{K}} \left\{ \eta \sum_{\tau=1}^{t-1} g_\tau(x) + \mathcal{R}(x) \right\},$$

where $\eta$ is a predefined learning rate, and $\mathcal{R}(x)$ is called a regularization function. Note that $\mathcal{R}(x)$ is chosen by the online player, and assumed to be $\sigma$-strongly convex and smooth, such that its second derivative is continuous.

Usually, general matrix norms are used to analyze and bound the regret of the RFTL algorithm: a PSD matrix $A \succ 0$ gives rise to the norm $\|x\|_A = \sqrt{x^\top A x}$; its dual norm is $\|x\|_{A^{-1}} = \|x\|_A^*$. In particular, the interesting case is when $A = \nabla^2 \mathcal{R}$, the Hessian of the regularization function. In this case, the notation is shorthanded to be $\|x\|_{\nabla^2 \mathcal{R}(y)} = \|x\|_y$ and $\|x\|_{\nabla^{-2} \mathcal{R}(y)} = \|x\|_y^*$.

Now, if we denote

$$\lambda = \sup_{t \in \{1,\dots,T\}, x,y \in \mathcal{K}} \left\{ \left( \|\nabla g_t(x)\|_y^* \right)^2 \right\} \quad \text{and} \quad R = \sup_{x,y \in \mathcal{K}} \left\{ \mathcal{R}(x) - \mathcal{R}(y) \right\},$$

then, the RFTL algorithm generates an online sequence $\{x_t\}_{t=1}^T$, for which the following holds:

$$R_T = \sum_{t=1}^T g_t(x_t) - \min_{x \in \mathcal{K}} \sum_{t=1}^T g_t(x) \le 2T\lambda\eta + \frac{R}{\eta}. \tag{5}$$

A complete analysis can be found in [2, 3].

### C.2 Adapting RFTL to the Framework of OCO with Memory

We start by defining the function $\tilde{f}_t$ as follows: $\tilde{f}_t(x) = f_t(x, \dots, x)$. Recall that $\tilde{f}_t(x)$ is convex in $x$ for all $t$, as assumed in Section 2. Following the notations of Section C.1, we define a regularization function $\mathcal{R}(x)$ and upper-bound

$$\lambda = \sup_{t \in \{1,\dots,T\}, x,y \in \mathcal{K}} \left\{ \left( \|\nabla \tilde{f}_t(x)\|_y^* \right)^2 \right\} \quad \text{and} \quad R = \sup_{x,y \in \mathcal{K}} \left\{ \mathcal{R}(x) - \mathcal{R}(y) \right\}. \tag{6}$$

Notice that $\lambda$ might depend implicitly on $m$. It follows that the loss functions $\{\tilde{f}_t\}_{t=1}^T$ are Lipschitz continuous for the Lipschitz constant $\sqrt{\lambda\sigma}$ with respect to the $\ell_2$-norm. I.e., it holds that

$$\left| \tilde{f}_t(x) - \tilde{f}_t(y) \right| \le \sqrt{\lambda\sigma} \cdot \|x - y\|.$$

The following is our main theorem, stated and proven:

**Theorem 3.1.** *Let $\{f_t\}_{t=1}^T$ be Lipschitz continuous loss functions with memory (from $\mathcal{K}^{m+1}$ to $[0,1]$), and let $R$ and $\lambda$ be as defined in Equation* (1)*. Then, Algorithm 1 generates an online sequence $\{x_t\}_{t=1}^T$, for which the following holds:*

$$R_{T,m} = \sum_{t=m}^T f_t(x_{t-m}, \dots, x_t) - \min_{x \in \mathcal{K}} \sum_{t=m}^T f_t(x, \dots, x) \le 2T\lambda\eta(m+1)^{3/2} + \frac{R}{\eta}.$$

*Setting $\eta = R^{1/2}(TL)^{-1/2}(m+1)^{-3/4}(\lambda/\sigma)^{-1/4}$ yields $R_{T,m} \le 3(TRL)^{1/2}(m+1)^{3/4}(\lambda/\sigma)^{1/4}$.*

*Proof.* First, note that applying Algorithm 1 to the loss functions $\{f_t\}_{t=1}^T$ is equivalent to applying the original RFTL algorithm to the loss functions $\{\tilde{f}_t\}_{t=1}^T$. I.e., given $m$ initial points $x_1, \dots, x_m$, both algorithms generate the same sequence of decisions $\{x_t\}_{t=m}^T$, for which it holds that:

$$\sum_{t=m}^T \tilde{f}_t(x_t) - \min_{x \in \mathcal{K}} \sum_{t=m}^T \tilde{f}_t(x) \le 2T\lambda\eta + \frac{R}{\eta},$$

or equivalently:

$$\sum_{t=m}^{T} f_t(x_t, \ldots, x_t) - \min_{x \in \mathcal{K}} \sum_{t=m}^{T} f_t(x, \ldots, x) \leq 2T\lambda\eta + \frac{R}{\eta}, \tag{7}$$

due to the regret guarantee in Equation (5). On the other hand, $f_t$ is Lipschitz continuous for the Lipschitz constant $L$, and thus we can bound

$$
\begin{aligned}
|f_t(x_t, \ldots, x_t) - f_t(x_{t-m}, \ldots, x_t)|^2 &\leq (L \cdot \|(x_t, \ldots, x_t) - (x_{t-m}, \ldots, x_t)\|)^2 \\
&= L^2 \cdot \sum_{j=1}^{m} \|x_t - x_{t-j}\|^2 \\
&\leq L^2 \cdot \sum_{j=1}^{m} \left( \sum_{l=1}^{j} \|x_{t-l+1} - x_{t-l}\| \right)^2 \\
&\leq L^2 \cdot \sum_{j=1}^{m} \left( \sum_{l=1}^{j} \frac{1}{\sqrt{\sigma}} \|x_{t-l+1} - x_{t-l}\|_{z_{t-l}} \right)^2 \\
&\leq L^2 \cdot \sum_{j=1}^{m} \left( \sum_{l=1}^{j} \frac{2\eta\sqrt{\lambda}}{\sqrt{\sigma}} \right)^2 \leq L^2 \cdot \sum_{j=1}^{m} \left( \frac{4m^2\eta^2\lambda}{\sigma} \right) \\
&\leq \frac{4L^2 m^3 \eta^2 \lambda}{\sigma},
\end{aligned}
$$

where $z_t \in [x_t, x_{t+1}]$. The inequality $\|x_{t+1} - x_t\|_{z_t} \leq 2\eta\sqrt{\lambda}$ follows from the standard analysis of the RFTL algorithm [2]. It follows that $|f_t(x_t, \ldots, x_t) - f_t(x_{t-m}, \ldots, x_t)| \leq 2L\eta m^{3/2}(\lambda/\sigma)^{1/2}$, and by summing over $t = m, \ldots, T$ we get that

$$\left| \sum_{t=m}^{T} f_t(x_t, \ldots, x_t) - \sum_{t=m}^{T} f_t(x_{t-m}, \ldots, x_t) \right| \leq 2TL\eta m^{3/2}(\lambda/\sigma)^{1/2}. \tag{8}$$

Next, by integrating Equations (7) and (8) we have that

$$R_{T,m} = \sum_{t=m}^{T} f_t(x_{t-m}, \ldots, x_t) - \min_{x \in \mathcal{K}} \sum_{t=m}^{T} f_t(x, \ldots, x) \leq 2TL\eta(m+1)^{3/2}(\lambda/\sigma)^{1/2} + \frac{R}{\eta}.$$

Finally, setting $\eta = R^{1/2}(TL)^{-1/2}(m+1)^{-3/4}(\lambda/\sigma)^{-1/4}$ yields the result stated in the theorem. $\square$

# D  Complete Analysis for Section 4

The outline of this section is as follows: we begin by adapting the EWOO algorithm of [23] to memoryless convex loss functions (Section D.1). Then, we present an algorithm for the standard OCO framework that attains low regret and small number of decision switches in expectation (Section D.2). Finally, we show that these properties together can be reduced to the framework of OCO with memory, yielding a nearly optimal policy regret bound (Section D.3).

## D.1  Adapting EWOO to Convex Loss Functions

Recall the Exponentially Weighted Online Optimization (EWOO) algorithm, presented in [23] and designed originally for $\alpha$-exp-concave (memoryless) loss functions $\{\ell_t\}_{t=1}^{T}$.

---

**Algorithm 5** Exponentially Weighted Online Optimization (EWOO)

---

1: Input: exp-concavity parameter $\alpha$, exp-concave loss functions $\{\ell_t\}_{t=1}^T$.
2: Initialize $w_1(x) = 1$ for all $x \in \mathcal{K}$, and choose $x_1 \in \mathcal{K}$ arbitrarily.
3: **for** $t = 1$ to $T$ **do**
4:     Play $x_t$ and suffer loss $\ell_t(x_t)$.
5:     Define weights $w_{t+1}(x) = e^{-\alpha \sum_{\tau=1}^t \ell_\tau(x)}$.
6:     Set $x_{t+1} = \left( \int_\mathcal{K} x \cdot w_{t+1}(x)dx \right) \cdot \left( \int_\mathcal{K} w_{t+1}(x)dx \right)^{-1}$
7: **end for**

$$R_T = \sum_{t=1}^T \ell_t(x_t) - \min_{x \in \mathcal{K}} \sum_{t=1}^T \ell_t(x) \le \frac{1}{\alpha} \left( 1 + n \log(T+1) \right).$$

Next, we consider the following modification of the EWOO algorithm — denoted as Algorithm 6.

---

**Algorithm 6**

---

1: Input: exp-concavity parameter $\alpha$, exp-concave loss functions $\{\ell_t\}_{t=1}^T$.
2: Initialize $w_1(x) = 1$ for all $x \in \mathcal{K}$, and choose $x_1 \in \mathcal{K}$ arbitrarily.
3: **for** $t = 1$ to $T$ **do**
4:     Play $x_t$ and suffer loss $\ell_t(x_t)$.
5:     Define weights $w_{t+1}(x) = e^{-\alpha \sum_{\tau=1}^t \ell_\tau(x)}$.
6:     Sample $x_{t+1}$ from the density function $p_t(x) = w_t(x) \cdot \left( \int_\mathcal{K} w_{t+1}(x)dx \right)^{-1}$
7: **end for**

---

Basically, $x_t$ is sampled from the density function $p_t(x) = w_t(x) \cdot \left( \int_\mathcal{K} w_t(x)dx \right)^{-1}$, instead of being computed deterministically. The following two lemmas state that applying Algorithm 6 to the loss functions $\{\hat{g}_t\}_{t=1}^T$ yields regret bound of $\mathcal{O}\big( \sqrt{T \log(T)} \big)$. We first bound the regret of Algorithm 6 when applied to general $\alpha$-exp-concave loss functions $\{\ell_t\}_{t=1}^T$ (Lemma D.1), and then plug in the loss functions $\{\hat{g}_t\}_{t=1}^T$ (Lemma D.2).

**Lemma D.1.** *Let $\{\ell_t\}_{t=1}^T$ be $\alpha$-exp-concave loss functions. Then, Algorithm 6 generates an online sequence $\{x_t\}_{t=1}^T$, for which the following holds:*

$$\mathbb{E}\left[ R_T \right] = \sum_{t=1}^T \mathbb{E}\left[ \ell_t(x_t) \right] - \min_x \sum_{t=1}^T \ell_t(x) \le \frac{1}{\alpha} \left( 1 + n \log(T+1) \right) + \frac{\alpha}{2} \sum_{t=1}^T \mathbb{E}\left[ \ell_t(x_t)^2 \right].$$

*Proof.* The proof goes along the lines of [23]; for completeness, we present here the full proof. Define $h_t(x) = e^{-\alpha \sum_{\tau=1}^{t-1} \ell_\tau(x)}$ and notice that

$$\mathbb{E}\left[ h_t(x_t) \right] = \int_\mathcal{K} h_t(x)p_t(x)dx = \frac{\int_\mathcal{K} \left( \prod_{\tau=1}^t h_\tau(x) \right) dx}{\int_\mathcal{K} \left( \prod_{\tau=1}^{t-1} h_\tau(x) \right) dx}.$$

Then, by telescopic product we have

$$\prod_{t=1}^T \mathbb{E}\left[ h_t(x_t) \right] = \frac{\int_\mathcal{K} \left( \prod_{t=1}^T h_t(x) \right) dx}{\int_\mathcal{K} 1 dx} = \frac{\int_\mathcal{K} \left( \prod_{t=1}^T h_t(x) \right) dx}{vol\left( \mathcal{K} \right)}, \tag{9}$$

where we used the fact that $w_1(x) = 1$ for all $x \in \mathcal{K}$. Denote $x^* = \arg\min_{x \in \mathcal{K}} \sum_{t=1}^T \ell_t(x)$, then it exists that $x^* = \arg\max_{x \in \mathcal{K}} \prod_{t=1}^T h_t(x)$. Define nearby points $\mathcal{S} \subset \mathcal{K}$ by

$$\mathcal{S} = \left\{ x \in \mathcal{K} \mid x = \frac{T}{T+1}x^* + \frac{1}{T}y \ , \ y \in \mathcal{K} \right\}.$$

By concavity and non-negativity of $h_t$ it holds that $h_t(x) \geq \frac{T}{T+1} h_t(x^*)$ for every $x \in \mathcal{S}$, and thus

$$\prod_{t=1}^{T} h_t(x) \geq \left(\frac{T}{T+1}\right)^T \prod_{t=1}^{T} h_t(x^*) \geq e^{-1} \prod_{t=1}^{T} h_t(x^*).$$

By substituting the above in Equation (9) and using the fact that $\mathcal{S}$ is a rescaling of $\mathcal{K}$ by factor of $\frac{1}{T+1}$ in $n$ dimensions, we have that

$$\prod_{t=1}^{T} \mathbb{E}\left[h_t(x_t)\right] = \frac{\int_{\mathcal{K}} \left(\prod_{t=1}^{T} h_t(x)\right) dx}{vol\,(\mathcal{K})} \geq \frac{\int_{\mathcal{S}} \left(\prod_{t=1}^{T} h_t(x)\right) dx}{vol\,(\mathcal{K})}$$

$$\geq \frac{\int_{\mathcal{S}} \left(e^{-1} \prod_{t=1}^{T} h_t(x^*)\right) dx}{vol\,(\mathcal{K})} = \frac{vol(\mathcal{S})}{vol\,(\mathcal{K})} e^{-1} \prod_{t=1}^{T} h_t(x^*)$$

$$= \frac{e^{-1}}{(T+1)^n} \prod_{t=1}^{T} h_t(x^*).$$

Now, by taking logarithm on both sides we get that

$$\sum_{t=1}^{T} \log\left(\mathbb{E}\left[h_t(x_t)\right]\right) - \sum_{t=1}^{T} \log\left(h_t(x^*)\right) \geq -1 - n \log(T+1),$$

or equivalently

$$\sum_{t=1}^{T} \log\left(\mathbb{E}\left[e^{-\alpha \ell_t(x_t)}\right]\right) + \alpha \sum_{t=1}^{T} \ell_t(x^*) \geq -1 - n \log(T+1). \tag{10}$$

Next, we use the facts that $e^{-x} \leq 1 - x + \frac{x^2}{2}$ for $0 \leq x \leq 1$ and $\log(1-x) \leq -x$ for $x < 1$, to derive the following inequality:

$$\log\left(\mathbb{E}\left[e^{-\alpha \ell_t(x_t)}\right]\right) \leq \log\left(\mathbb{E}\left[1 - \alpha \ell_t(x_t) + \frac{\alpha^2}{2} \ell_t(x_t)^2\right]\right)$$

$$= \log\left(1 - \alpha \mathbb{E}\left[\ell_t(x_t)\right] + \frac{\alpha^2}{2} \mathbb{E}\left[\ell_t(x_t)^2\right]\right)$$

$$\leq -\alpha \mathbb{E}\left[\ell_t(x_t)\right] + \frac{\alpha^2}{2} \mathbb{E}\left[\ell_t(x_t)^2\right]$$

By substituting the above in Equation (10) and rearranging we get that

$$\sum_{t=1}^{T} \mathbb{E}\left[\ell_t(x_t)\right] - \sum_{t=1}^{T} \ell_t(x^*) \leq \frac{1}{\alpha}\left(1 + n \log(T+1)\right) + \frac{\alpha}{2} \sum_{t=1}^{T} \mathbb{E}\left[\ell_t(x_t)^2\right],$$

as stated in the lemma. $\qquad \square$

Plugging in the loss functions $\{\hat{g}_t\}_{t=1}^{T}$ into the previous lemma yields the following result:

**Lemma D.2.** *Let $\{g_t\}_{t=1}^{T}$ be convex functions from $\mathcal{K}$ to $[0,1]$, such that $D = \sup_{x,y \in \mathcal{K}} \|x - y\|$ and $G = \sup_{x,t} \|\nabla g_t(x)\|$, and define $\hat{g}_t(x) = g_t(x) + \frac{\eta}{2}\|x\|^2$ for some $\eta \leq \frac{G}{D}$. Then, Applying Algorithm 6 to the loss functions $\{\hat{g}_t\}_{t=1}^{T}$ generates an online sequence $\{x_t\}_{t=1}^{T}$, for which the following holds:*

$$\mathbb{E}\left[R_T\right] = \sum_{t=1}^{T} \mathbb{E}\left[g_t(x_t)\right] - \min_{x} \sum_{t=1}^{T} g_t(x) \leq \frac{4G^2}{\eta}\left(1 + n \log(T+1)\right) + \frac{T\eta}{2}\left(\frac{\left(1 + \eta D^2\right)^2}{4G^2} + D^2\right).$$

*Setting $\eta = \frac{2G}{D}\sqrt{\frac{1 + \log(T+1)}{T}}$ yields $\mathbb{E}\left[R_T\right] \leq 8n \cdot \max\left\{GD, \frac{1}{GD}\right\} \cdot \sqrt{T(1 + \log(T+1))}$.*

*Proof.* Recall that the loss functions $\{\hat{g}_t\}_{t=1}^T$ are $\frac{\eta}{4G^2}$-exp-concave for $\eta \le \frac{G}{D}$. Thus, applying Algorithm 6 to the loss functions $\{\hat{g}_t\}_{t=1}^T$ yields the following result (using Lemma D.1):

$$\sum_{t=1}^T \mathbb{E}\left[\hat{g}_t(x_t)\right] - \min_x \sum_{t=1}^T \hat{g}_t(x) \le \frac{4G^2}{\eta}\left(1 + n\log(T+1)\right) + \frac{\eta}{8G^2}\sum_{t=1}^T \mathbb{E}\left[\hat{g}_t(x_t)^2\right].$$

By substituting $\hat{g}_t$ from the definition and using the fact that $\hat{g}_t(x) \in [0, 1 + \eta D^2]$ for all $t$ and $x \in \mathcal{K}$, we have that

$$\sum_{t=1}^T \mathbb{E}\left[g_t(x_t)\right] - \min_x \sum_{t=1}^T g_t(x)$$

$$\le \frac{4G^2}{\eta}\left(1 + n\log(T+1)\right) + \frac{\eta}{2}\sum_{t=1}^T \left(\frac{\left(1 + \eta D^2\right)^2}{4G^2} + \|x^*\|^2 - \|x_t\|^2\right).$$

The lemma is obtained by observing that $\|x^*\|^2 - \|x_t\|^2 \le D^2$. $\qquad\square$

## D.2 Algorithm and Analysis

We turn now to restate and prove our main theorem:

**Theorem 4.1.** *Let $\{g_t\}_{t=1}^T$ be convex functions from $\mathcal{K}$ to $[0, 1]$, such that $D = \sup_{x,y \in \mathcal{K}} \|x - y\|$ and $G = \sup_{x,t} \|\nabla g_t(x)\|$, and define $\hat{g}_t(x) = g_t(x) + \frac{\eta}{2}\|x\|^2$ for some $\eta \le \frac{G}{D}$. Then, Algorithm 2 generates an online sequence $\{x_t\}_{t=1}^T$, for which it holds that*

$$\mathbb{E}\left[R_T\right] = \sum_{t=1}^T \mathbb{E}\left[g_t(x_t)\right] - \min_{x \in \mathcal{K}} \sum_{t=1}^T g_t(x) \le \frac{4G^2}{\eta}\left(1 + n\log(T+1)\right) + \frac{T\eta}{2}\left(\frac{\left(1 + \eta D^2\right)^2}{4G^2} + D^2\right),$$

*and in addition*

$$\mathbb{E}\left[S\right] = \mathbb{E}\left[\sum_{t=1}^T \mathbb{1}_{\{x_{t+1} \neq x_t\}}\right] \le \frac{T\eta}{4G^2} + \frac{TD^2\eta^2}{8G^2},$$

*Setting $\eta = \frac{2G}{D}\sqrt{\frac{1 + \log(T+1)}{T}}$ yields $\mathbb{E}\left[R_T\right] = \mathcal{O}\left(\sqrt{T\log(T)}\right)$, and $\mathbb{E}\left[S\right] = \mathcal{O}\left(\sqrt{T\log(T)}\right)$.*

*Proof.* The proof follows immediately by observing that: (1) Algorithm 2 generates the decisions from the same distribution with respect Algorithm 6 (stated formally in Lemma D.3 below), and thus attains the same expected regret bound; and (2) Algorithm 2 has an expected low switches guarantee (also stated below in Lemma D.4). $\qquad\square$

We shall continue to prove the lemmas.

**Lemma D.3.** *Let $\{g_t\}_{t=1}^T$ be convex functions from $\mathcal{K}$ to $[0, 1]$, such that $D = \sup_{x,y \in \mathcal{K}} \|x - y\|$ and $G = \sup_{x,t} \|\nabla g_t(x)\|$, and define $\hat{g}_t(x) = g_t(x) + \frac{\eta}{2}\|x\|^2$ for some $\eta \le \frac{G}{D}$. Denote by $\{y_t\}_{t=1}^T$ and $\{x_t\}_{t=1}^T$ the online sequences generated by applying Algorithm 2 and Algorithm 6 to the loss functions $\{g_t\}_{t=1}^T$ and $\{\hat{g}_t\}_{t=1}^T$, respectively. Then, it holds that $y_t$ and $x_t$ are sampled from the same distribution for all $t$.*

*Proof.* Let $q_t(\cdot)$ and $p_t(\cdot)$ be the density functions of $y_t$ and $x_t$, respectively, and $W_t = \int_{\mathcal{K}} w_t(x)dx$. The proof is by induction: for $t = 1$ we have from the definition that $p_1(x) = q_1(x)$ for all $x \in \mathcal{K}$. Now, let us assume that $p_{t-1}(x) = q_{t-1}(x)$ for all $x \in \mathcal{K}$, and prove for $t$. Notice that the weights

update for both algorithms is the same and is independent of the decisions actually played by the player. Thus, by applying the law of total probability we have that

$$
\begin{aligned}
q_t(x) &= p_{t-1}(x) \cdot \frac{w_t(x)}{w_{t-1}(x)} + p_t(x) \cdot \int_{\mathcal{K}} p_{t-1}(y) \left( 1 - \frac{w_t(y)}{w_{t-1}(y)} \right) dy \\
&= \frac{w_{t-1}(x)}{W_{t-1}} \cdot \frac{w_t(x)}{w_{t-1}(x)} + \frac{w_t(x)}{W_t} \cdot \int_{\mathcal{K}} \frac{w_{t-1}(y)}{W_{t-1}} \left( \frac{w_{t-1}(y) - w_t(y)}{w_{t-1}(y)} \right) dy \\
&= \frac{w_t(x)}{W_{t-1}} + \frac{w_t(x)}{W_t} \cdot \int_{\mathcal{K}} \frac{w_{t-1}(y) - w_t(y)}{W_{t-1}} dy \\
&= \frac{w_t(x)}{W_{t-1}} + \frac{w_t(x)}{W_t} \cdot \frac{W_{t-1} - W_t}{W_{t-1}} \\
&= \frac{w_t(x) \cdot W_t + w_t(x) \cdot W_{t-1} - w_t(x) \cdot W_t}{W_{t-1} \cdot W_t} \\
&= \frac{w_t(x) \cdot W_{t-1}}{W_{t-1} \cdot W_t} = \frac{w_t(x)}{W_t} = p_t(x).
\end{aligned}
$$

The above holds for all $x \in \mathcal{K}$, and thus the lemma is obtained. □

**Lemma D.4.** *Let $\{g_t\}_{t=1}^T$ be convex functions from $\mathcal{K}$ to $[0,1]$, such that $D = \sup_{x,y \in \mathcal{K}} \|x - y\|$ and $G = \sup_{x,t} \|\nabla g_t(x)\|$. Then, applying Algorithm 2 to the loss functions $\{g_t\}_{t=1}^T$ generates an online sequence $\{x_t\}_{t=1}^T$, for which the it holds that*

$$
\mathbb{E}[S] = \sum_{t=1}^T \mathbb{E}\left[ 1_{\{x_{t+1} \neq x_t\}} \right] \leq \frac{T\eta}{4G^2} + \frac{TD^2\eta^2}{8G^2},
$$

*where $S$ denotes the number of decision switches in the sequence $\{x_t\}_{t=1}^T$.*

*Setting $\eta = \frac{2G}{D}\sqrt{\frac{1+\log(T+1)}{T}}$ yields $\mathbb{E}[S] \leq 1 + \log(T+1) + \frac{1}{GD}\sqrt{T(1+\log(T+1))}$.*

*Proof.* From Algorithm 2 it follows that

$$
\mathbb{E}\left[ 1_{\{x_{t+1} \neq x_t\}} \right] = P\left( x_{t+1} \neq x_t \right) \leq 1 - \frac{w_{t+1}(x_t)}{w_t(x_t)} = 1 - e^{-\frac{\eta}{4G^2}\hat{g}_t(x_t)},
$$

Using the inequality $1 - e^{-x} \leq x$ for all $x$, and substituting $\hat{g}_t$ from the definition yields

$$
1 - e^{-\frac{\eta}{4G^2}\hat{g}_t(x_t)} \leq \frac{\eta}{4G^2} g_t(x_t) + \frac{\eta^2}{8G^2} \|x_t\|^2.
$$

Next, by summing the above for all $t$ we have that

$$
\sum_{t=1}^T \mathbb{E}\left[ 1_{\{x_{t+1} \neq x_t\}} \right] \leq \frac{\eta}{4G^2} \sum_{t=1}^T g_t(x_t) + \frac{\eta^2}{8G^2} \sum_{t=1}^T \|x_t\|^2.
$$

Finally, since $\|x\|^2 \leq D^2$ for all $x \in \mathcal{K}$ and $g_t(x) \in [0,1]$ for all $x \in \mathcal{K}$ and $t \in \{1, \ldots, T\}$, setting $\eta = \frac{2G}{D}\sqrt{\frac{1+\log(T+1)}{T}}$ gives the stated result. □

### D.3 Adaptation to the Framework of OCO with Memory

Up to this point, we presented an algorithm that attains $\mathcal{O}(\sqrt{T \log(T)})$-regret along with expected $\mathcal{O}(\sqrt{T \log(T)})$ decision switches for generally convex loss functions $\{g_t\}_{t=1}^T$. The next lemma states that these two properties imply learning against bounded-memory adversaries.

**Lemma D.5.** *Let $\{f_t\}_{t=1}^T$ be loss functions with memory from $\mathcal{K}^{m+1}$ to $[0,1]$, define $\tilde{f}_t(x) = f_t(x,\ldots,x)$, and denote $D = \sup_{x,y\in\mathcal{K}} \|x - y\|$ and $G = \sup_{x,t} \|\nabla \tilde{f}_t(x)\|$. Then, applying Algorithm 2 to the loss functions $\{\tilde{f}_t\}_{t=1}^T$ yields an online sequence $\{x_t\}_{t=1}^T$, for which it holds that:*

$$\mathbb{E}\left[R_{T,m}\right] = \sum_{t=1}^T \mathbb{E}\left[f_t(x_{t-m},\ldots,x_t)\right] - \min_{x\in\mathcal{K}} \sum_{t=1}^T f_t(x,\ldots,x)$$

$$\leq \frac{4G^2}{\eta}\left(1 + n\log(T+1)\right) + \frac{T\eta}{2}\left(\frac{\left(1+\eta D^2\right)^2}{4G^2} + D^2\right) + \frac{Tm\eta}{4G^2} + \frac{TD^2 m\eta^2}{8G^2}.$$

*Setting $\eta = \frac{2G}{D}\sqrt{\frac{1+\log(T+1)}{mT}}$ yields $\mathbb{E}\left[R_{T,m}\right] \leq 8n \cdot \max\left\{GD, \frac{1}{GD}\right\} \cdot \sqrt{mT(1 + \log(T+1))}.$*

*Proof.* From Theorem 4.1, we know that applying Algorithm 2 to the loss functions $\{\tilde{f}_t\}_{t=1}^T$ yields:

$$\sum_{t=1}^T \mathbb{E}[\tilde{f}_t(x_t)] - \min_{x\in\mathcal{K}} \sum_{t=1}^T \tilde{f}_t(x) \leq \frac{4G^2}{\eta}\left(1 + n\log(T+1)\right) + \frac{T\eta}{2}\left(\frac{\left(1+\eta D^2\right)^2}{4G^2} + D^2\right),$$

or equivalently:

$$\sum_{t=1}^T \mathbb{E}\left[f_t(x_t,\ldots,x_t)\right] - \min_{x\in\mathcal{K}} \sum_{t=1}^T f_t(x,\ldots,x)$$

$$\leq \frac{4G^2}{\eta}\left(1 + n\log(T+1)\right) + \frac{T\eta}{2}\left(\frac{\left(1+\eta D^2\right)^2}{4G^2} + D^2\right). \tag{11}$$

Now, notice that if a decision switch did not occur between rounds $(t - m)$ and $t$, it trivially holds that $f_t(x_{t-m},\ldots,x_t) = f_t(x_t,\ldots,x_t)$. Otherwise, if a decision switch did occur between these rounds, we can bound $|f_t(x_{t-m},\ldots,x_t) = f_t(x_t,\ldots,x_t)| \leq 1$. Thus, it follows that

$$\sum_{t=m}^T |f_t(x_{t-m},\ldots,x_t) - f_t(x_t,\ldots,x_t)| \leq m \cdot S,$$

where again, $S$ denotes the number of decision switches in the sequence $\{x_t\}_{t=1}^T$. From Lemma D.4 we have that $\mathbb{E}\left[S\right] \leq \frac{T\eta}{4G^2} + \frac{TD^2\eta^2}{8G^2}$, and it follows that

$$\left|\sum_{t=m}^T \mathbb{E}\left[f_t(x_{t-m},\ldots,x_t)\right] - \sum_{t=m}^T \mathbb{E}\left[f_t(x_t,\ldots,x_t)\right]\right| \leq \sum_{t=1}^T |\mathbb{E}\left[f_t(x_{t-m},\ldots,x_t) - f_t(x_t,\ldots,x_t)\right]|$$

$$\leq \sum_{t=1}^T \mathbb{E}\left[|f_t(x_{t-m},\ldots,x_t) - f_t(x_t,\ldots,x_t)|\right]$$

$$\leq m \cdot \mathbb{E}\left[S\right] \leq \frac{Tm\eta}{4G^2} + \frac{TD^2 m\eta^2}{8G^2}.$$

Plugging the above in Equation (11) yields the result stated in the lemma. □

# E  Efficient Implementation of Algorithm 2

The original EWOO algorithm (Algorithm 5) of [23] is not efficient, since it generates $x_t$ as the expectation with respect to the distribution $p_t$ in every round. *Hazan et al.* solve this issue by referring to the works of [24], that offer a sampling method from logconcave distributions. These techniques enable the sampling of $m$ points from the distribution $p_t$ in time of $\tilde{\mathcal{O}}(n^4 + mn^3)$. Since an accuracy of $T^{-1}$ to the expectation is necessary for maintaining logarithmic regret, $m$ must be on the order of $T^2$. Thus, generating a single decision $x_t$ via a slightly modified EWOO algorithm requires running time of $\tilde{\mathcal{O}}(n^4 + T^2 n^3)$, which results in a total running time of $\tilde{\mathcal{O}}(Tn^4 + T^3 n^3)$.

The implementation of the proposed algorithm (Algorithm 2) can rely on the same techniques as algorithm EWOO, yet can be carried out more efficiently in various ways. First, our algorithm requires only $\tilde{\mathcal{O}}(T^{1/2})$ samples (in compare to $T$ samples that EWOO requires), due to its low switches guarantee. Second, each of these samples requires time of $\tilde{\mathcal{O}}(n^4)$ using the techniques of [24], because $x_t$ need not be generated as the expectation of $p_t$, but rather only be sampled from this distribution. Therefore, an efficient implementation of our algorithm can be carried out in a total running time of $\tilde{\mathcal{O}}(T^{1/2}n^4)$.

Another efficient implementation of Algorithm 2 relies on the work of [25], in which techniques of random walks are utilized for regret minimization. Basically, these techniques are applicable in our setting for two reasons: (1) two successive distributions over the decision set, $p_t$ and $p_{t+1}$, are relatively close; and (2) each distribution $p_t$ can be approximated quite well using a Gaussian distribution. This allows sampling $x_{t+1}$ via a random walk technique that requires only one step, due to the fact that $x_t$ can be used as its warm start. This results in a same running time guarantee for our algorithm, as stated before for the techniques of [24].

## Footnotes

[5]We refer the reader to [20, 21] for more comprehensive information about **OLS** and **Johansen**.

[23] prove the following regret bound for Algorithm 5: