[Reviews · NeurIPS 2015]

Submitted by Assigned_Reviewer_1

The paper investigates Online Convex Optimization (OCO) with bounded memory loss functions (i.e. the loss incurred depends on the last few actions taken). The regret is measured compared to the best fixed action in hindsight. Earlier works have established bounds for the experts and bandits version of this problem. This work generalizes to OCO. Two algorithms are proposed. One based on Follow-The-Regularized-Leader, and one based on Shrinking Dartboard. For the analysis of FTRL a Lipschitz condition is essential. As vanilla FTRL does not have large parameter movement, the effect of simply ignoring the memory is small. For the exponential weights version it is instead exploited that Shrinking Dartboard only uses few parameter updates. The regret and run-time of both algorithms are discussed.

The first algorithm is FTRL on all fixed actions. The crux of the main result here is to bound the actual loss in terms of the loss of some fixed action. In line 185 it is claimed that this is easy using the Lipschitz assumption. The problem is that the Lipschitz constant $\lambda$ is computed purely based on $\tilde f$. In line 710 it is claimed that the former can be enlarged without loss of generality. Enlarging $\lambda$ is fine, but then this should percolate to the main Theorem 3.1. One may think that this is purely a matter of constants, but I am not so sure. In particular, I worry that the factor necessary to correctly upgrade $\lambda$ may depend (as mentioned in line 702) on the memory length $m$ (for natural families of functions parametrized by the memory length $m$), which is the main dependence investigated in this paper.

I am confused that you think of past-action dependent losses as a property ("adaptive") of the adversary. The fact that the loss functions have memory is merely a feature of the protocol specification. We might just as well encounter such past-dependent-losses in stochastic scenarios. Past-action-dependent losses are all around (switching cost, transaction cost, ...). Whether the same can be said about bounded-memory adaptive adversaries is a matter of metaphysics.

Line 025: theoretical

Line 045: The examples given for dependence on past actions are the standard examples for log loss. What do these have to do with past-action dependent losses? For example when compressing a file, where is the adaptive adversary? Are you worried about rounding bits? Arithmetic coding is the solution.

Line 053: "that is, must" -> "that is, it must"

Line 071: results

Section 2.2. Please introduce $m$.

Line 147: The counterfactualism is not as heavy as you say. The algorithms only use the loss of each fixed action.

Line 147: "We assume ... revealed ... aware". No need to assume, this is a consequence of revelation.

Line 186: "algorithm requires". Neither $\lambda$ nor $R$ occur in the algorithm.

Line 223: the ratio of weights equals $e^{-\alpha \hat g_t(x_t)}$. Is that not simpler?

Theorem 4.1. So Algorithm 2 is not anytime?

Line 239: $y_{t-j}$ is missing a transpose.

Line 304: This holds with equality.

Line 315: Algorithm 3.

Line 340: Please point to the run-time discussion in the supplementary material.

Line 498: Is this a standard data set? Where can we find it?

**** UPDATE ****

I read the author response and reread sections 3 and appendix C, and updated my score accordingly. As the authors indicate, Theorem 3.1 can be amended to incorporate the Lipschitz dependence explicitly. This will not characterize the memory dependence for reasonable loss function classes, but the authors say that this is not their focus and I can live with that.

With this out of the way, I now see a paper presenting reasonable progress on an interesting problem, and I no longer oppose it being accepted.
Summary: An interesting problem. Unfortunately, the main theorem is problematic.

Submitted by Assigned_Reviewer_2

The paper extends the notion of online learning with adversarial memory to the OCO framework. The authors propose two efficient algorithms: first one applies to (strongly) convex and Lipschitz functions while the second applies to convex functions, without requiring Lipschitz property. The authors complement their results with experiments.

Originality- I will not question this aspect of the paper. The authors mostly adapt existing theory in online learning but do it in a clever way, which, if correct (I will come to this later) is worthy of acceptance. Significance- I believe the idea will be interesting to the online learning community.

Clarity- I have some minor issues regarding the clarity of the paper. I will list them so that (hopefully) the authors find it easier to read. 1. A concrete example of a function which calculates loss based on last $m$ moves of learner would have been helpful in the introduction.The authors simply say "However, in coding, compression, portfolio selection and more, the adversary is not completely memoryless". While I am sure I can find examples in the references, the introduction should be more enticing and the paper self-containing. 2. Line 156- It is mentioned that f_t is convex if $\tilde{f}(x)= f(x,x,...,x)$ is convex. Considering $f_t$ is a multivariate function, should it not be that $f_t$ being convex naturally mean that it is jointly convex in all $m$ arguments? Otherwise, why claim $f_t$ to be convex? 3. Theorem 3.1- The regret is for Lipschitz continuous function with constant L (line 176). However, L does not even appear in the regret equation. Looking at the proof, it seems L has somehow been absorbed in \lambda and \sigma. This is very confusing to someone reading the main paper.

And in line 705, why is the Lipschitz constant $\sqrt(\lambda) \sigma$? 4. Though I understand the more involved work is for (non)-Lipschitz functions, Algorithm 1. and its' proof is extremely simple when regularizer is $\ell_2$ norm (basically OGD). In fact, in OGD, two consecutive iterates $x_{t+1}$ and $x_t$ are close as long as the gradient is controlled (it is when the function is Lipschitz). Maybe the analysis of Alg.1 can be moved to the supplement with more discussion about Alg 2. in the main paper. 5. Why are \hat{g}_t exp-concave with \eta/4G^2 (line 893)? I understand its exp-concave but why with factor \eta/4G^2?

Quality- My major concern is with the proof of one lemma. I am enlisting my concern here: 1. Lemma D.3- The crucial part of Alg.2 is the bound on the number of switches (As formally stated in Lemma D.4).

So effectively, the proof goes through the following line of argument: Alg.5 is modified in Alg.6, which changes the way $x_{t+1}$ is sampled. Through a series of arguments, it comes to the inequality in Lemma D.2. Then Theorem 4.1 follows by using a bound on number of switches.

My concern is with Lemma D.3. First of all, both Alg.2 and Alg.6 are being applied to exp-concave function $\hat{g}_t$ (Change the function $\ell_t$ to $\hat{g}_t$ in Alg.6). I dont think it is correct to claim Alg.2 is being applied to function $g_t$. The loss is being calculated in terms of $g_t$ but the update is based on $\hat{g}_t$, since $\hat{g}_t$ is exp-concave. So the only difference between Alg.2 and Alg.6 is how the sequence $x_t$ is being generated.

Crucially, in Alg.6, $x_t$ is being sampled from $p_t(x)$. In Alg.2 (using the same exp-concave functions), $x_t$ is being generated from $p_t(x)$ w.p.

1- (w_t(x_{t-1})/w_{t-1}(x_{t-1})) and $x_t=x_{t-1}$

with complementary probability. Since $p_t$ is same density function in Alg.2 and Alg.6, how can the two sequences come

from same distribution? I find that very much counter-intuitive, and this lies in the heart of analysis of Alg.2. Moreover, when i tried to go through the proof of Lemma D.3, i got stuck at the first line. Why is Line 947 true?

Overall, I am not sure about the veracity of the claim. I am of course not claiming it is wrong but I think its crucial to verify the proof. I would request other reviewers/ PC members to check the proof of Lemma D.3 at least.

Update: After discussion, I have understood the proof of Lemma D.3. I have updated by rating. Assuming that the results have not appeared anywhere before, the paper should be accepted
Summary: While I generally liked the paper and I think it can definitely be a good paper (given correctness) worthy of a top conference after some polish, I have some major issue with one of the proofs and a few minor issues regarding the general presentation of the paper. My ratings are tentative and I will surely reconsider after author feedback and discussions.

Author Feedback
Author rebuttal: We thank the reviewers for their time and comments.

Before we answer to specific comments, we address a common concern to some of the reviewers: the issue of L not appearing in the result of Theorem 3.1.

In the proof of the theorem we assume without loss of generality that L \leq \sqrt{ \sigma \lambda} merely in order to clean the result and statements (this can be done since \lambda is an upper bound on the gradient norms and thus can be enlarged to satisfy this condition). The result and the proof techniques would still hold without this assumption, and the alternative regret bound will be:

(TRL)^{1/2} (\lambda / \sigma)^{1/4} (m+1)^{3/4}

As we state in line 702, \lambda can implicitly depend on m, but since our paper focuses on the question of whether the difficulty introduced by the memory affects the optimality of the online learner w.r.t. T, we find this fact less crucial. Moreover, one cannot hope to have a constant value of \lambda in general, since the argument of f_t is of dimension m \times n. This means that the value of \lambda should be computed ad-hoc according to the problem at hand. For example, in both of our applications \lambda is independent of m.

Reviewer 1
We address your concern about \lambda above.

Your second concern: "I am confused that you think of past-action dependent losses as a property ("adaptive") of the adversary" is a misinterpretation of line 49: "an important aspect of our work is that the memory is \emph{not} used to relax the adaptiveness of the adversary (cf. [5, 6]), but rather to model the feedback received by the player". We agree with you that the fact that the loss functions have memory is merely a feature of the protocol specification: the whole paper is written accordingly. We hope that this also resolves the examples issue.

Line 147: your observation about the counterfactual feedback is indeed correct, but notice however that this only strengthens our result.
Line 186: R and \lambda are needed to determine the value of \eta.
Line 223: we find both equally simple.
Line 304: the inequality is in fact necessary as X need not be a rank 1 matrix.
Line 498: the dataset is publicly available (we downloaded it from Yahoo! Finance).

We thank you for the other comments, which will be fixed if the paper is accepted.

Reviewer 3:
1. We will certainly incorporate an example in the introduction.
2. Line 156: we introduce here a new notion of convexity: convexity with memory. This is the minimal assumption we could find on f_t in order to still be able to learn efficiently. What you are referring to is the standard notion of convexity, which is too strict in our setting. In particular, this notion does not hold for our first application.
3. We answered this concern in our opening paragraph of this rebuttal.
4. We agree with you that the discussion on Alg. 2 should be expanded, but we cannot see a way to do it on the expense of Section 3. We will try to elaborate on it instead of something else.
5. We will add a citation to this fact, which can be found for example in: Bubeck, Sebastien. "Introduction to online optimization." Lecture Notes(2011).

Reviewer 2:
We actually do not use any smoothness conditions in the paper, and we further state that we are not using any blocking technique (line 64).

Reviewer 4:
Thank you for your concise review. Multistep ahead prediction can in fact be done using other delayed feedback algorithms, and this is exactly our point: the framework of OCO with memory captures the delayed feedback as a special case. Moreover, our algorithm for this type of problems turns out to be much simpler than existing algorithms. We will elaborate more on this issue.

Reviewer 5:
We presented two algorithms in the paper and not one (it seems from your writing that you missed the second one). We hope to answer your concern about the Lipschitz constant in the opening paragraph of this rebuttal.

Reviewer 6:
We answered your concerns about the dependence of the regret bound on m in the opening paragraph of this rebuttal.